# The Role of CAR-T Cells in Peritoneal Carcinomatosis from Gastric Cancer: Rationale, Experimental Work, and Clinical Applications

**DOI:** 10.3390/jcm10215050

**Published:** 2021-10-28

**Authors:** Siyuan Qian, Pedro Villarejo-Campos, Damián García-Olmo

**Affiliations:** 1Department of Surgery, Fundación Jimenez Diaz University Hospital, Avda. Reyes Católicos, 2, 28040 Madrid, Spain; siyuan.qianv@quironsalud.es (S.Q.); damian.garcia@uam.es (D.G.-O.); 2Department of Surgery, Universidad Autónoma de Madrid, C/ Arzobispo Morcillo s/n, 28034 Madrid, Spain

**Keywords:** gastric adenocarcinoma, peritoneal carcinomatosis, intraperitoneal chemotherapy, cytoreductive surgery, immunotherapy, cell therapy

## Abstract

Cytoreductive surgery (CRS) and hyperthermic intraperitoneal chemotherapy (HIPEC) have shown poor effectiveness in treating peritoneal carcinomatosis (PC) of gastric origin with a high tumor burden (high peritoneal cancer index), though there are scarce therapy alternatives that are able to improve survival. In experimental studies, chimeric antigen receptor-T (CAR-T) cell therapy has shown encouraging results in gastric cancer and is currently being evaluated in several clinical trials. Regarding PC, CAR-T cell therapy has also proven useful in experimental studies, especially when administered intraperitoneally, as this route improves cell distribution and lifespan. Although these results need to be supported by ongoing clinical trials, CAR-T cells are a promising new therapeutic approach to peritoneal metastases from gastric cancer. In this review, we summarize the current evidence of the use of CAR-T cells in gastric cancer and PC of gastric origin.

## 1. Introduction

Gastric cancer (GC) is the fifth most frequently diagnosed cancer and the third leading cause of cancer-related death. More than one million new cases are diagnosed every year worldwide [1]. The worldwide 5-year overall survival rate is between 20% and 40%, and the median overall survival rarely reaches 12 months [2,3,4]. Despite recent advances in treatment, more than 50% of patients who undergo surgery for GC relapse locally or develop systemic metastases. Moreover, one half of patients are diagnosed with synchronic systemic metastases associated with a 5-year survival rate below 10% [4].

The peritoneum is a common site of metastasis, and peritoneal metastases are associated with low rates of patient survival [4]. Following complete cytoreduction, the 5-year overall survival rate in patients with gastric peritoneal carcinomatosis (PC) is 13% to 23% [5]. Half of the patients with GC will develop PC, and PC will be present in up to 20% of patients with a potential resectable tumor. Furthermore, after surgical treatment, the peritoneum can be the only site of recurrence in up to 34% of cases [6].

Conventional treatment with chemotherapy, radiotherapy, or surgery is the therapy of choice in most cancer patients [7]. However, the efficacy of these strategies is limited in advanced GC due to its genetic complexity and heterogeneity [8]; thus, there is an urgent need to develop precise, personalized therapeutic approaches [9]. This need is even more pressing in patients with PC from GC. The current treatment, consisting of cytoreductive surgery combined with hyperthermic intraperitoneal chemotherapy (HIPEC), offers a median overall survival of 18 to 21 months [10,11,12].

Currently, advances in immunotherapy have enabled the clinical use of monoclonal antibodies. Human epidermal growth factor receptor 2 (HER2) is commonly used as a target in immunotherapy, as it is overexpressed in 10% to 20% of GC cases. Trastuzumab is a monoclonal antibody that targets this receptor and is mainly used in breast and gastric cancer, improving patient survival [13,14].

The use of chimeric antigen receptor-T (CAR-T) cells is a new type of immunotherapy developed over recent decades and consists of modifying patients’ own T-lymphocytes to attack a specific target. By means of genetic engineering using viral vectors, the CAR is introduced into the T cell, which enables the cell to recognize a selected tumor-associated antigen (TAA) in a major histocompatibility class-independent manner [15].

Over the last 5 years, four types of CAR-T cells targeting CD19 have shown promising results in hematological malignancies and have since been approved by the FDA: axicabtagene ciloleucel (trade name, Yescarta), tisagenlecleucel (trade name, Kymriah), lisocabtagene maraleucel (trade name, Breyanzi), and brexucabtagene autoleucel (trade name, Tecartus) [16,17,18,19].

### 1.1. Chimeric Antigen Receptor-T (CAR-T) Cell Therapy

The development of CAR-T cells began in the late 1980s. Early investigations resulted in the first generation of cells which had a structure that consisted of a single chain variable fragment (ScFv) region and the CD3ζ intracellular domain. These cells were unsuccessful in clinical trials as they were able to activate but did not proliferate, which indicated low efficacy [20,21,22,23]. The second and third generation of CAR-T cells included one or two costimulatory molecules, respectively, in their intracellular domain (Figure 1) [24,25]. These costimulatory molecules improved the efficacy and persistence of CAR-T cells [23,24,26,27,28,29]. 

However, tumors can escape from CAR-T cell activity due to their immunosuppressive microenvironment, which inhibits CAR-T function. As a consequence, the fourth generation of CAR-T cells, or TRUCKs (T cells redirected for universal cytokine-mediated killing), are currently being designed. These are capable of secreting proinflammatory cytokines that can block the inhibitory effect of the tumor microenvironment, thereby enhancing their antitumoral activity in an immunosuppressive microenvironment (Figure 2) [23,24,25].

Selecting an optimal target is essential to the success of CAR-T therapy. The right target must meet two requirements: Firstly, the target should be expressed in all tumor cells; if not, antigen-negative tumor cells can escape the action of CAR-T cells and result in tumor recurrence. Secondly, the target should be present only in tumor cells and be absent from healthy cells in order to avoid an on-target, off-tumor response in which CAR-T cells attack non-tumor cells [30].

CD19 in hematologic malignancies (e.g., B-cell lymphoma, childhood acute lymphoblastic leukemia, adult-onset ALL) meets these criteria. CAR-T cells targeting CD19 have achieved a complete response in 70% to 90% of patients [31,32,33,34]. These encouraging results led to research into the application of this type of immunotherapy in solid tumors with the hope of reproducing the same result as in hematologic malignancies.

However, solid tumors have some peculiarities that hematologic malignancies do not. Firstly, TAAs are more heterogeneous in solid tumors, and finding an antigen that fits with the target selection criteria is difficult [35]. Secondly, it is difficult for CAR-T cells to migrate and penetrate the tumor [36]. Lastly, CAR-T cells encounter a hostile tumor microenvironment that averts the action of these cells; this physical barrier consists of stroma and an immunologic barrier composed of immunosuppressive cells and metabolites [30,37].

### 1.2. Adverse Effects Associated with CAR-T Cell Therapy

Cytokine-release syndrome is the most common side effect in patients undergoing CAR-T cell therapy and is sometimes associated with fatal outcomes. Between 50% and 90% of patients who receive anti-CD19 CAR-T cell therapy could develop cytokine-release syndrome during the first week after infusion. This effect is related to treatment response and tumor burden. Cytokine-release syndrome results from immunologic over-activation caused by CAR-T cells, which receive an exaggerated signal owing to receptor stimulation. This produces cytokine release, activating myeloid cells (mainly monocytes and macrophages) that generate a systemic inflammatory response mediated by IL-6 and IL-1. The symptoms of cytokine-release syndrome vary widely. General malaise and nausea are the most frequent symptoms, though the first symptom to present is often fever. Nonetheless, the disease may progress clinically to acute respiratory distress, acute renal failure, disseminated intravascular coagulation, cardiomyopathy, or even arrhythmia [9].

CAR-T cell-related encephalopathy or immune effector cell-associated neurotoxicity syndrome (ICANS): Between 30% and 90% of patients who receive anti-CD19 CAR-T cell therapy develop neurotoxicity manifesting as mild confusion or, in severe cases, cerebral edema [9]. This syndrome is related to a systemic inflammatory response induced by myeloid cells that activate endothelial cells producing von Willebrand factor and Ang-2 that promote blood–brain barrier dysfunction.

On-target, off-tumor response: Caused by the reaction of the immune system, this response triggers the activation of CAR-T cells against healthy tissues expressing the target antigen. This response is more common in solid tumors. The intensity and frequency of this response vary according to the target and route of administration used. To avoid it, a highly specific receptor for the tumor target must be selected, CAR-T cells must have high affinity and specificity for the receptor chosen, and the CAR-T cells must be administered at an appropriate dose [30].

A potential solution for decreasing the severity of the adverse effect is the local administration of CAR-T cells, i.e intratumoral or intraperitoneal. The peritoneal route increases the local concentration of effector cells, which triggers a local immune response in the peritoneal cavity and minimizes the adverse systemic effects. This lower systemic toxicity is related to the binding of CAR-T cells and tumor cells on the peritoneal surface (extravascular tissue) which involves limited cytokine release into the bloodstream. Furthermore, we hypothesize that the systemic release of cytokines is hindered by the existence of a fibrous stroma with a collagen-rich extracellular matrix, characteristic of peritoneal metastases.

## 2. Rationale

The peritoneal surface is a frequent site of involvement in GC, conferring poor prognosis. Therapeutic approaches in patients with peritoneal metastases of gastric origin and high tumor burden have shown poor results [38].

Multiple studies have explored the role of CAR-T in gastrointestinal cancers, obtaining promising results regarding their capacity to eliminate tumor cells and their metastases. However, most are preclinical studies in animal models. The choice of an experimental in vivo model of peritoneal carcinomatosis to evaluate the efficacy and potential side effects of CAR-T therapy is critical before initiating a phase I/II clinical trial. Xenograft, humanized, syngeneic or transgenic are available murine in vivo models for CAR-T cell therapy research.

CAR-T cell therapies designed for a number of different targets in GC have proven useful in experimental studies and currently are being evaluated in registered clinical trials with advanced GC patients.

### 2.1. Safety and Efficacy of CAR-T Therapy for Advanced Gastric Cancer: Ongoing Clinical Trials

The main trials related to CAR-T cell therapies in GC with available data are shown in Table 1.

### 2.2. Carcinoembryonic Antigen (CEA)

Carcinoembryonic antigen (CEA) is a glycoprotein related to GC and other gastrointestinal tumors. In clinical practice it is used to estimate the severity of gastrointestinal cancer and to detect recurrence among patients in follow-up. This antigen is an attractive target for CAR-T cell therapy as it is not detected in healthy adult tissues and is only expressed on the luminal face of gastrointestinal and lung cells. However, tumor cells lose this polarity and express CEA on their entire surface. Since normal cells express CEA on the luminal side of the cell, they remain invisible to immune cells. This attribute enables CAR-T cells to distinguish normal cells from tumor cells [39]. When administered in the hepatic artery, CAR-T cells that target CEA have been found to be safe in patients with liver metastases, causing no severe adverse effects [40].

Another phase I clinical trial studied the efficacy of second-generation anti-CEA CAR-T cells administered systemically in escalating doses to treat colorectal liver metastasis. The authors of the study demonstrated that these cells are well-tolerated, even at high dose levels, with no toxic effects [41].

Although the clinical efficacy of first-generation carcinoembryonic antigen (CEACAM5)-specific CAR-T cells was limited due to respiratory toxicity and short persistence [20], new generations of CEA CAR-T cells are currently being evaluated in GC patients.

### 2.3. HER2 (Human Epidermal Growth Factor Receptor 2)

HER2, or ERBB2, is an antigen present on the cell surface. When activated, this antigen promotes cell proliferation and inhibits apoptosis. Amplification or overexpression of HER2 has been associated with several cancers, including GC. Approximately 10% to 30% of GCs have HER2 amplification; however, this characteristic confers a poor prognosis in terms of tumor growth, lymph-node involvement, and metastases [42,43]. Thus, targeting HER2 in patients with HER2 amplification could be an optimal therapeutic approach. Trastuzumab is a monoclonal antibody used in GC, and when used in combination with chemotherapy in patients with HER2 overexpression, it has a demonstrated benefit for overall survival [44]. Nevertheless, as some patients develop resistance to this drug, new strategies are needed, such as CAR-T cells that target HER2.

A preclinical study in a murine model by Han et al. demonstrated that systemic HER2 CAR-T cell therapy was effective in treating HER2-positive GC and did not affect tissues expressing low levels of HER2, thus suggesting no on-target, off-tumor effect. Furthermore, the authors observed Tcell persistence suggestive of protection against recurrence [45].

### 2.4. ICAM (Intercellular Adhesion Molecule 1)

ICAM, another cell surface glycoprotein, is involved in cell––cell and cell–extracellular matrix adhesion. ICAM is overexpressed in 40% of GC patients and is related to a worse prognosis. ICAM is found more frequently in advanced stages, such as lymph-node metastases and systemic metastases [46]. In mice, the use of ICAM in CAR-T cell therapy has shown a strong capacity for tumor elimination and can also act on distant tumors [47].

### 2.5. Claudin 18.2 (CLDN18.2)

CLDN18.2 is a stomach-specific isoform of claudin-18, and plays an important role in cell junctions. It is highly expressed in healthy gastric cells, but only within the differentiated epithelial cells of gastric mucosa, and is also expressed in gastric tumor cells, their metastases, and in pancreatic and in esophageal adenocarcinoma [48]. 

Jiang et al. showed that CLDN18.2 CAR-T could produce a complete response in a xenograft murine model. Furthermore, despite the fact that healthy cells express CLDN18.2, no obvious toxicities were found [49]. Note that patients with GC undergo total gastrectomy as part of their treatment, thus on-target, off-tumor responses should be avoided by removing the stomach. 

A phase I pilot study used CAR-T anti-CLDN18.2 in seven patients with gastric cancer and five with pancreatic cancer. One gastric patient achieved a complete response, and there were two partial responses. No severe adverse effects were observed, indicating that this treatment is safe and well-tolerated [50].

### 2.6. EpCAM

Epithelial cell adhesion molecule (EpCAM) is a transmembrane glycoprotein expressed on epithelial cells and is involved in cell–cell adhesion as well as signaling, migration, proliferation, and cell differentiation. EpCAM is overexpressed in more than 90% of GCs and is associated with a poor prognosis. Clinical trials with EpCAM as a target for CAR-T cell therapy in advanced gastric cancer are currently underway [51].

Meanwhile, other targets for CAR-T therapy in gastric cancer studied in clinical trials include mesothelin, MUC1, CD276, CD44v6 (transmembrane glycoproteins), and ROR2 (tyrosine-protein kinase transmembrane receptor) (Table 1).

### 2.7. CAR-T in Peritoneal Carcinomatosis of Gastric Origin

The peritoneal surface is frequently involved in GC, conferring a poor prognosis, and peritoneal disease will develop in up to 50% of patients with advanced GC [6]. Currently, the most studied and validated prognostic factor is the peritoneal carcinomatosis index (PCI). The PCI quantifies the extent of tumor spread on the peritoneal surface. A PCI of less than 7 is an independent prognostic factor associated with higher overall survival in peritoneal carcinomatosis of gastric origin [52,53].

The discovery and development of new biomarkers, such as exosomes produced by tumor cells which are detectable in ascitic fluid or circulating blood, can help us to choose the appropriate therapeutic strategy for each patient. New nanotechnologies, such as templated plasmonic exosomes (TPEX), make it possible to detect exosomes in patients’ body fluids [54]. Measurement of exosomes in the ascitic fluid of patients with gastric peritoneal carcinomatosis may be useful in identifying patients who are non-responders to intraperitoneal chemotherapy [55].

These patients would be suitable candidates for new target therapies such as CAR-T cells. As mentioned previously, multiple studies have explored the role of CAR-T in GC, obtaining promising results in terms of their capacity to eliminate tumor cells and their metastases. Currently, there is an urgent need for new therapeutic approaches to treat gastric PC, and CAR-T cells could be a potential treatment.

## 3. Experimental Works and Clinical Applications

### 3.1. Route of CAR-T Administration Cells in PC

Treatment of PC using CAR-T was first described by Katz et al., who used CAR-T cells targeting CEA to treat colorectal PC in an animal model. The authors observed that intraperitoneal delivery was superior to systemic administration. 

Intraperitoneal infusion was associated with higher tumor reduction and a more durable effect compared to systemic administration, thus suggesting a protection against recurrence and against other distant metastases [56].

Another group designed a second-generation CAR-T cell strategy targeting TAG72 in a murine model with ovarian PC. Intraperitoneal administration was superior to intravenous administration, showing increased antitumor activity and better overall survival, which was further enhanced with repeated infusions [57].

Ang et al. created a murine model of PC and developed CAR-T cells against EpCAM by means of mRNA transfection. This type of transfection resulted in temporary action, which offers a higher safety in case of adverse effects. Due to the transient effect, repeated infusions are needed for an optimal effect [58].

Therefore, these results suggest that local administration allows better CAR-T cell trafficking and infiltration, greater antitumor activity, and added protection against recurrence and extraperitoneal antitumor effect while decreasing systemic adverse effects. However, there are still challenges inherent to solid tumors and PC that must be overcome. The most important hurdle is the tumor microenvironment, which confers a physical barrier and an immunosuppressive microenvironment. One of the physical barrier compounds is the tumor stroma, which contain high levels of collagen in the extracellular matrix. This stroma prevents systemic and local treatments from reaching the tumor cells. However, this collagen can be destroyed using collagenase, therefore facilitating drug penetration [59].

Furthermore, intraperitoneal administration of CAR-T cells has shown antitumor activity at distant sites such as subcutaneous nodules. This is not a result of the direct action of CAR-T cells; rather, it is the consequence of an effect similar to the abscopal effect that occurs in radiotherapy [60]. This mechanism is triggered by the release of multiple tumor antigens after being destroyed by CAR-T cells, allowing them to be recognized by dendritic cells and creating an immune response against antigens other than the CAR-T target [60].

### 3.2. CAR-T in Gastric PC

Though there are several studies investigating the usefulness of CAR-T in PC, studies focused on peritoneal carcinomatosis from GC are scarce.

Third-generation CAR-T cells targeting mesothelin as treatment for GC and PC in a murine model showed that intravenous administration could produce tumor regression and even elimination, and reports have shown persistence of CAR-T cells in peripheral blood after 2 weeks. This resulted in prolonged survival in mice treated with CAR-T compared with no CAR-T. The researchers also observed that peritumoral injection in subcutaneous implants significantly reduced tumor growth compared with intravenous administration [61]. Likewise, intraperitoneal HER2 CAR-T cells in PC from GC demonstrated higher efficacy with prolonged mice survival and significantly delayed tumor growth compared to administrating untransduced T cells intraperitoneally [45]. Another group targeted ICAM-1 and compared intraperitoneal injection versus tail vein injection. They observed a significantly higher tumor response with intraperitoneal delivery [47].

The CAR-T cell therapy approach in PC originating from GC is being evaluated in two phase I clinical trials (Table 2). The targets used in these trials are CEA and EpCAM, which are suitable receptors for intraperitoneal administration because both are specifically expressed on the entire surface of peritoneal tumor cells, which facilitates direct contact with CAR-T cells.

## 4. Conclusions

CAR-T cell therapies have been widely investigated and have achieved extraordinary results in hematological malignancies. However, the application of this approach in solid tumors has fallen short of expectations, and its efficacy is still unclear. Solid tumors contain certain hurdles not present in hematological malignancies, which explains the lack of efficacy of CAR-T cell therapy in these tumors. The main obstacle is related to the tumor microenvironment, which provides the tumor with a physical barrier and plays an immunosuppressive role. Many studies are evaluating new types of CAR-T cells and optimizing them for greater efficacy, persistence, and migration. In addition to modifying CAR-T cells to improve their action, these cells are also being combined with other therapies such as monoclonal antibodies, chemotherapy, collagenase, or surgery, which may improve their results.

The use of CAR-T cells in GC and PC appears to be safe and feasible. The results of preclinical studies with these cells are promising and several targets may be used. Additionally, since patients with GC undergo gastrectomy, specific antigens present in healthy gastric cells can be used as targets without an on-target, off-tumor response, thereby expanding the number of potential targets.

Intraperitoneal delivery of CAR-T cells is the best route of administration as it promotes migration to, and infiltration of, the tumor cells, as well as averting the systemic toxicity of the microenvironment. Despite the existence of ongoing clinical trials needed to confirm the results of preclinical studies, regional CAR-T cell therapy for the treatment of PC provides promising evidence for its use and a new hope in the search for a treatment for this lethal disease. The possible selection of patients with gastric peritoneal carcinomatosis for CAR-T cell therapy is shown in a flowchart (Figure 3).

## Figures and Tables

**Figure 1 jcm-10-05050-f001:**
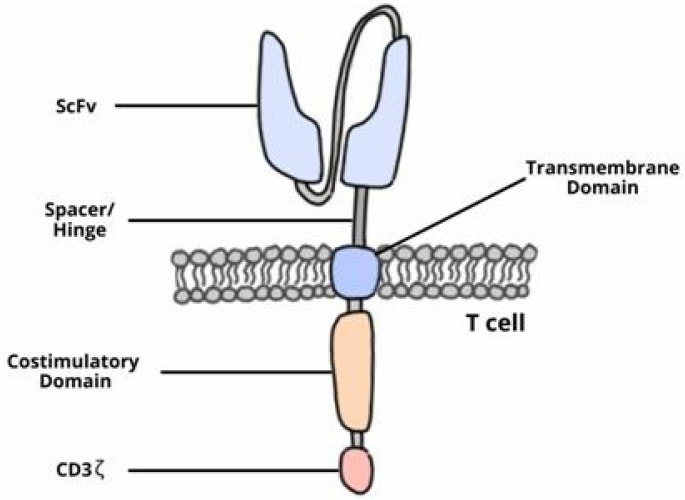
The CAR-T structure is composed of ScFv: This region enables the T cell to recognize the target antigen; hinge or spacer: binds the ScFv to the transmembrane domain and contributes to ScFv flexibility; transmembrane domain: the link between the extracellular and intracellular regions; CD3z: an intracellular signaling domain that activates the T cell; and costimulatory molecules: involved in improving CAR-T proliferation and persistence.

**Figure 2 jcm-10-05050-f002:**
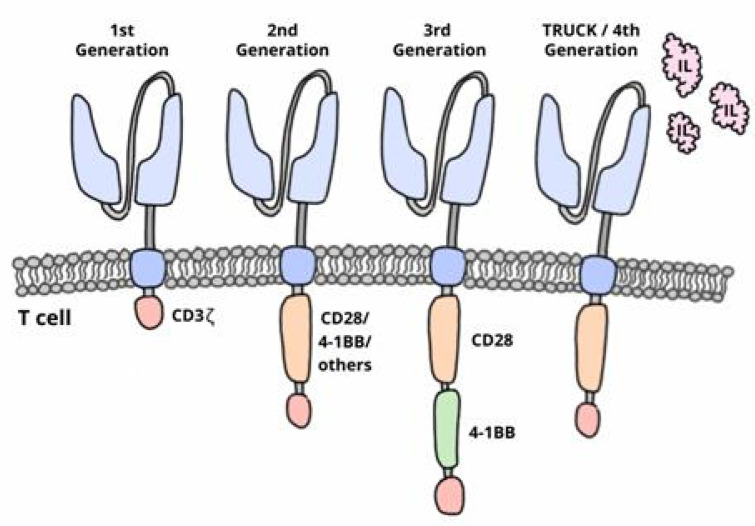
Generations of CAR-T cells.

**Figure 3 jcm-10-05050-f003:**
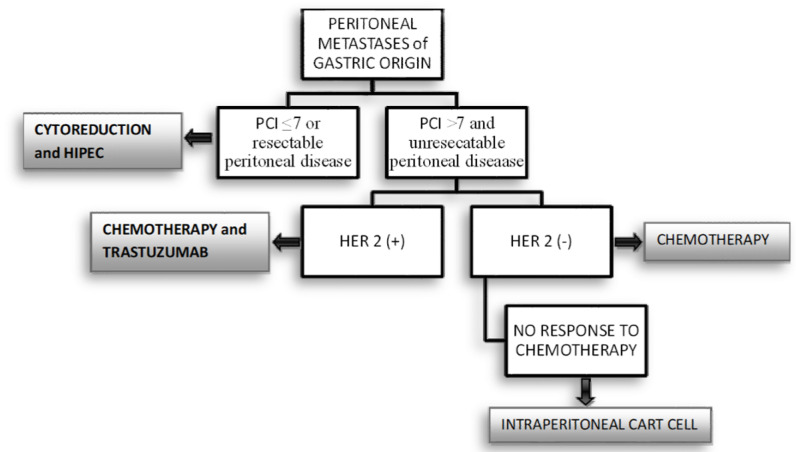
CAR-T cell therapy flowchart.

**Table 1 jcm-10-05050-t001:** Ongoing clinical trials with CAR-T cell therapy in gastric cancer (source: clinicaltrials.gov (accessed on 25 October 2021)).

Target Antigen	Identifier	Gastric Cancer Condition	Trial Phase	CAR-T Infusion Route	Start-Completion Date	Study Status	Country
EpCAM	NCT02725125	EpCAM-positive relapsed or refractory advanced gastric adenocarcinoma	I/II	Intravenous	November 2015–November 2019	Unknown	China
	NCT03013712	EpCAM-positive relapsed or refractory advanced gastric adenocarcinoma	I/II	Intravenous	January 2017–December 2020	Unknown	China
Claudin18.2	NCT04467853	Claudin18.2-positive advanced gastric adenocarcinoma	I	Intravenous	September 2020–November 2024	Recruiting	China
	NCT03890198	Claudin18.2-positive advanced gastric adenocarcinoma	I	Intravenous	April 2019–March 2020	Early discontinuation	China
	NCT04404595	Claudin18.2-positive advanced gastric adenocarcinoma	I	Intravenous	October 2020–September 2035	Recruiting	United States
	NCT04581473	Claudin18.2-positive advanced gastric adenocarcinoma	I/II	Intravenous	October 2020–December 2022	Recruiting	China
HER2	NCT04650451	HER2-positive advanced or metastatic gastric adenocarcinoma	I/II	Intravenous	December 2020–January 2025	Recruiting	United States
	NCT04511871	HER2-positive relapsed or refractory advanced gastric adenocarcinoma	I	Intravenous	July 2020–January 2023	Recruiting	China
	NCT02713984	HER2-positive relapsed or refractory advanced gastric adenocarcinoma	I/II	Intravenous	March 2019–July 2019	Withdrawn	China
	NCT03740256	HER2-positive advanced or metastatic gastric adenocarcinoma	I	Intravenousin combination with intratumorCAdVEC (oncolytic adenovirus)	June 2021–December 2038	Recruiting	United States
Mesothelin	NCT03941626	Mesothelin-positive advanced gastric adenocarcinoma, unresectable or refractory to chemoradiotherapy	I/II	intravenous	September 2019–December 2021	Recruiting	China
	NCT03638206	Mesothelin-positive relapsed or refractory advanced gastric adenocarcinoma	I/II	intravenous	Marah 2018–March 2023	Recruiting	China
	NCT04348643	CEA-positive relapsed or refractory advanced gastric adenocarcinoma	I/II	intravenous	February 2020–April 2023	Recruiting	China
CEA	NCT02349724	CEA-positive relapsed or refractory advanced gastric adenocarcinoma	I	intravenous	December 2014–December 2019	Unknown	China
MUC1	NCT02617134	MUC1-positive relapsed or refractory advanced gastric adenocarcinoma	I/II	intravenous	November 2015–November 2018	Unknown	China
CD276	NCT04864821	CD276-positive advanced gastric adenocarcinoma	I	Intravenous and intratumor	May 2021–May 2023	Recruitment not begun	China
ROR2	NCT03960060	ROR2-positive relapsed or refractory advanced gastric adenocarcinoma	I	intravenous	May 2019–June 2023	Recruitment closed	China
CD44v6	NCT04427449	CD44v6-positive advanced gastric adenocarcinoma	I/II	Intravenous	June 2020–December 2023	Recruiting	China

**Table 2 jcm-10-05050-t002:** Clinical trials with CAR-T cell therapy in peritoneal carcinomatosis of gastric origin (source: ClinicalTrials.gov (accessed on 25 October 2021)).

Identifier	Trial Phase	Target Antigen	Gastric Cancer Condition	CAR-T Infusion Route	Start-Completion Date	Study Status	Country
NCT03563326	I	EpCAM	Peritoneal metastasis	Intraperitoneal	Aug 2018-Dec 2022	Recruiting	China
NCT03682744	I	CEA	Peritoneal metastases or malignant ascites	Intraperitoneal	Sep 2018-Mar 2021	Active, recruitment closed	United States

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
