# Peer review of "The Role of CAR-T Cells in Peritoneal Carcinomatosis from Gastric Cancer: Rationale, Experimental Work, and Clinical Applications"

_jcm, 2021, doi:10.3390/jcm10215050_

Round 1

Reviewer 1 Report

This is an interesting review highlighting the promise of CART-T cells in PC of gastric origin.  Although the use of CART-T cell therapy is not developed in the field of PC, this review is useful as it informs clinicians/surgeons working in this field of a potential utility of CART-T cell therapy in a disease with limited options

Suggestions to improve the manuscript

1)Highlight the side effects of CART-T cell therapy as a separate section and why IP use may mitigate/reduce the risk of cytokine storm

2)Highlight the use of potential biomarkers which may be used to select patients for targeted therapy.  Eg. work by the Shao group in the use of biomarkers for peritoneal carcinomatosis (PC) have been shown to be prognostic and potentially predict response to therapy.  This aspect needs to be highlighted (Dual-Selective Magnetic Analysis of Extracellular Vesicle Glycans, Exosome-templated nanoplasmonics for multiparametric molecular profiling) It is unlikely that the use of CART-T cell therapy will be useful for all PC without stratifying the patients appropriately

3)Highlight the need to develop good in-vivo models of PC for testing esp in the field of utilising CART-T cell therapy

Minor points--an overview picture of how CART-T cell therapy in a flow diagram/summary will be useful at the end of the review to help readers understand this field easily

a)Identification of antigens--generation of CART-T cell--experimental validation--early phase trial followed by RCT

Author Response

Thank you very much for your suggestions and for your helpful comments. We have followed your valuable recommendations and have replied to your comments below.

1)Highlight the side effects of CART-T cell therapy as a separate section and why IP use may mitigate/reduce the risk of cytokine storm

Thanks for your suggestion, we have added these sections to our manuscript.

  • Adverse effects associated with CAR-T cell therapy

Cytokine-release syndrome (CRS) is the most common side effect in patients undergoing CAR-T cell therapy and is sometimes associated with fatal outcomes. Between 50% and 90% of patients who receive anti-CD19 CAR-T cell therapy could develop it, during the first week after the infusion. This effect is related to treatment response and tumor burden. CRS results from an immunologic over-activation caused by CAR-T cells, which receive an exaggerated signal owing to receptor stimulation. This produces cytokine release, activating myeloid cells (mainly monocytes and macrophages) that generate a systemic inflammatory response mediated by IL-6 and IL-1. The symptoms of CRS vary widely. General malaise and nausea are the most frequent symptoms, though the first symptom to present is often fever. Nonetheless, the disease may progress clinically to acute respiratory distress, acute renal failure, disseminated intravascular coagulation, cardiomyopathy, or even arrhythmia.

CAR-T cell-related encephalopathy or immune effector cell-associated neurotoxicity syndrome (ICANS): Between 30% and 90% of patients who receive anti-CD19 CAR-T cell therapy develop neurotoxicity manifesting as mild confusion or, in severe cases, cerebral edema. This syndrome is related to a systemic inflammatory response induced by myeloid cells that activate endothelial cells producing von Willebrand factor and Ang-2 that promoting blood-brain barrier dysfunction.

On-target, off-tumor response: Caused by the reaction of the immune system, this response triggers activation of CAR T cells against healthy tissues expressing the target antigen. This response is more common in solid tumors. The intensity and frequency of this response vary according to the target and route of administration used. To avoid it, a highly specific receptor for the tumor target must be selected, CAR T cells must have high affinity and specificity for the receptor chosen, and the CAR T cells must be administered at an appropriate dose.

The peritoneal route increases the local concentration of effector cells, which triggers a local immune response in the peritoneal cavity and minimizes the adverse systemic ef-fects. This lower systemic toxicity is because contact between CART cells and tumor cells occurs on the peritoneal surface (extravascular tissue) and the eventual release of cytokines into the bloodstream is very limited. Furthermore, we hypothesize that the arrival of cytokines to the systemic circulation is hindered by the existence of a fibrous stroma with a collagen-rich extracellular matrix, characteristic of peritoneal metastases.

2)Highlight the use of potential biomarkers which may be used to select patients for targeted therapy.  Eg. work by the Shao group in the use of biomarkers for peritoneal carcinomatosis (PC) have been shown to be prognostic and potentially predict response to therapy.  This aspect needs to be highlighted (Dual-Selective Magnetic Analysis of Extracellular Vesicle Glycans, Exosome-templated nanoplasmonics for multiparametric molecular profiling) It is unlikely that the use of CART-T cell therapy will be useful for all PC without stratifying the patients appropriately

Thank you again for your very helpful comment, we have added the following paragraphs to improve the manuscript

The peritoneal surface is frequently involved in GC, conferring poor prognosis and synchronous peritoneal disease in GC is present in 20% of cases. Currently, the most studied and validated prognostic factor is the peritoneal carcinomatosis index (PCI). PCI quantifies the extent of tumor spread on the peritoneal surface. PCI less than 7 is an independent prognostic factor associated with overall survival in peritoneal carcinomatosis of gastric origin.

The discovery and development of new biomarkers, such as exosomes produced by tumor cells and detectable in ascitic fluid or circulating blood, can help us to choose the appropriate therapeutic strategy for each patient. New nanotechnologies, such as templated plasmonic exosomes (TPEX), make it possible in patients' body fluids. Measurement of exosomes in the ascitic fluid of patients with gastric peritoneal carcinomatosis may be useful in to identify patients who are non-responders to intraperitoneal chemotherapy.

These patients and would be suitable candidates for new target therapies such as CAR-T cells.

3)Highlight the need to develop good in-vivo models of PC for testing esp in the field of utilising CART-T cell therapy

The choice of an experimental in vivo model of peritoneal carcinomatosis to evaluate the efficacy and potential side effects of CAR-T therapy is critical before initiating a phase I/II clinical trial. Xenograft, humanized, syngeneic or transgenic are useful murine in vivo models for CAR-T cell therapy research.

Minor points--an overview picture of how CART-T cell therapy in a flow diagram/summary will be useful at the end of the review to help readers understand this field easily

We have added a diagram to clarify for readers the possible applications of CART cell therapy in peritoneal gastric carcinomatosis (see Figure 3: CAR-T cell therapy flowchart)

a)Identification of antigens--generation of CART-T cell--experimental validation--early phase trial followed by RCT

The section "Safety and efficacy of CAR-T therapy for advanced gastric cancer: ongoing clinical trials" describes the main antigens with experimental validation.

The tables added to the manuscript (Table 1 and Table 2) identify the antigens used in each early phase of clinical trials.

Unfortunately, it has been impossible to identify the different CAR-T cell generations for each clinical trial, because this information is not described in several protocols.

Reviewer 2 Report

The authors summarized and suggested CAR-T cell therapy for gastric cancer patients with peritoneal metastasis. They suggested intraperitoneal CAR-T therapy with some evidence from other solid cancers and in vivo experiments. Clinically gastric cancer with peritoneal mets is challenging, thus important, and CAR-T cell therapy is one of promising next-step treatment in oncology tough some issues to be adapted in solid cancer. (as described by the authors)

The evidence for CAR-T cell therapy for gastric cancer peritoneal metastasis is still immature, but this review summarized well about the current evidence for it.  

Author Response

Thank you very much for your valuable comment, we are very grateful for your appreciation.